# Preference-based Antibody Expression Ranking: Scaling with Large-scale Weak Supervision

**Josh Qixuan Sun** [1 2]   **Morteza Babaie** [1]   **Wenyang Hou** [1]   **Mark Crowley** [2]   **David Young** [1]

## Abstract

Antibody expression ranking is a critical task in antibody design, yet its modelling is severely hindered by the scarcity of labeled experimental data. To address this, we propose a unified preference-based learning framework that integrates scarce quantitative expression data with large-scale weak positive supervision from immunization data. We adapt Direct Preference Optimization (DPO) to protein language models by introducing a union-masked log-likelihood approximation and IMGT-based alignment, enabling efficient training on variable-length sequences. Evaluating on a diverse internal dataset of 1254 labeled sequences and 4 million unlabeled camelid-derived antibodies, we show that our method consistently outperforms baselines on most metrics. Our results demonstrate that preference learning can effectively learn from weak supervision, providing a scalable solution for antibody expressibility optimization in data-constrained settings. Project page: https://kisoji-biotechnology-inc.github.io/Preference-Expression-Ranking/.

## 1. Introduction

Antibody expression ranking is a critical bottleneck in antibody development, yet the biophysical principles governing why some sequences achieve high yields while others fail remain poorly understood. This unpredictability motivates the use of computational models to guide candidate selection. In practical antibody campaigns, enrichment and AI-based generation routinely produce hundreds to thousands of candidates, so the relevant objective is not precise yield prediction for individual sequences, but a ranked prioritization of candidates with the highest likelihood of expressibility. Realizing this objective, however, is challenging due to the extreme scarcity of labeled experimental data; public benchmarks typically contain only hundreds of sequences with limited diversity in antibody length and yield distributions, failing to reflect the complexity of real design (Table 1).

Simultaneously, industrial antibody discovery workflows generate additional sources of supervision that are largely absent from existing benchmarks. Standard immunization-based discovery pipelines routinely produce millions of unique antibody sequences derived from immune responses (Hanke et al., 2020; Tsuruta et al., 2024). Although these sequences are not associated with quantitative expression measurements, empirical evidence from internal experimental studies suggests that more than 90% of camelid-derived antibodies are expressible, whereas the expression rate drops to around 60% for antibodies derived from transgenic mice or generated by AI models. This provides a form of weak positive supervision, an informative but underutilized signal that lies between supervised and unsupervised learning.

In this work, we investigate the integration of scarce, quantitative expression measurements with large-scale weak positive supervision for antibody optimization. We utilize Exp-1K, a proprietary dataset of 1254 sequences that, unlike existing benchmarks (Table 1), includes both expressible and non-expressible examples with substantial length variability. To complement this dataset, we leverage another private dataset, Camel-4M, a collection of four million camelid-derived antibody sequences obtained from immunization, which we treat as a source of weak positive supervision.

The central challenge is to design a training framework that can naturally absorb these heterogeneous supervision signals. We address this challenge through Direct Preference Optimization (DPO) (Rafailov et al., 2023). DPO provides a unified abstraction for antibody expression ranking: quantitative expression yields can be converted into relative preferences between sequences, while weak positive supervision induces implicit preferences over the sequence space without requiring explicit labels. Under this formulation, a single model can jointly learn to rank sequences by relative yield score and discriminate expressible from non-expressible antibodies.

[1]Kisoji Biotechnology Inc, Montreal, Quebec, Canada. [2]Department of Electrical and Computer Engineering, University of Waterloo, Waterloo, Ontario, Canada. Correspondence to: Josh Qixuan Sun <josh.q.sun@uwaterloo.ca>.

*Proceedings of the $43^{rd}$ International Conference on Machine Learning*, Seoul, South Korea. PMLR 306, 2026. Copyright 2026 by the author(s).

| Dataset | $N$ | Yield Range (mg/L) | Non-exp | Avg. Len. | Len. Std. | Source |
|---|---|---|---|---|---|---|
| Garbinski et al. (2023) | 94 | [68.3, 373.3] | $\times$ | 122.6 | 2.2 | (Chungyoun et al., 2024), originally unpublished |
| Jain et al. (2017) | 136 | [6.6, 277.2] | $\times$ | 119.5 | 3.2 | (Jain et al., 2017) |
| Szkodny et al. (2024) | 178 | [1.5, 22.8] | $\times$ | 120.0 | 0.0 | (Szkodny & Lee, 2024) |
| PROPHET-Ab | 246 | [34.3, 781.9] | $\times$ | 149.0 | 0.0 | (Arsiwala et al., 2025) |
| Exp-1K (Ours) | 1254 | [0.0, 435.5] | $\checkmark$ | 123.3 | 12.8 | Internal |
| Camel-4M (Ours) | 4.2M | - | - | 119.7 | 4.6 | Internal |

*Table 1.* Comparison of antibody expression benchmarks. Our Exp-1K dataset significantly exceeds existing benchmarks in both scale and coverage, notably including non-expressible sequences and exhibiting higher sequence length variability. Camel-4M provides an additional large-scale source of weak positive supervision derived from immunization.

Our framework begins with a domain-alignment phase, where the model undergoes continual pretraining with masked language modeling on immunization-derived Camel-4M to adapt existing protein language models (pLMs) to specific protein families (Alley et al., 2019; Biswas et al., 2021). This establishes a specialized representation of the antibody space, providing a robust foundation for preference learning. Building on this initialization, we introduce a modified DPO objective to pLMs. Applying DPO to pLMs is non-trivial, as pLMs are mostly masked language models and the sequence likelihood is defined differently from causal language models. Specifically, pLMs rely on pseudo-log-likelihood (Salazar et al., 2020; Meier et al., 2021), which is computationally expensive when applied independently at each sequence position. We address this limitation by adopting a union-masked log-likelihood approximation (Ferragu et al., 2025), where differing positions between two sequences are jointly masked and evaluated in a single forward pass. To enable consistent construction of such preference pairs for sequences with variable lengths, we align antibody sequences with IMGT numbering (Lefranc et al., 2003a;b) using ANARCI software (Dunbar & Deane, 2016). This design allows efficient and scalable preference optimization for antibody sequences under realistic length variability.

Empirically, we evaluate our framework on the Exp-1K dataset, demonstrating that preference-based learning effectively captures antibody expressibility. Our results show that our model consistently outperforms standard regression and classification baselines in both yield ranking and binary prediction. Specifically, we observe that the combined effect of domain-aligned MLM and preference optimization leads to substantial performance gains compared to training on labeled data alone. Furthermore, we show that our framework scales effectively with increasing amounts of weak supervision, confirming that it can successfully extract useful biophysical signals from large-scale, unlabeled antibody sequences. These findings suggest that preference-based optimization is a practical and robust approach for antibody design in data-constrained scenarios.

We summarize our primary contributions as follows: (a) Unified Framework: We develop a preference-based learning framework that integrates scarce quantitative yields and large-scale weak positive supervision within a single training objective. (b) Efficient DPO for pLMs: We adapt a union-masked log-likelihood approximation and the IMGT-based alignment, enabling efficient training on variable-length sequences. (c) Empirical Validation: We demonstrate that our approach consistently outperforms baselines with multiple protein language backbones. (d) Scaling Analysis: We show that our framework effectively scales with increasing volumes of weak supervision, proving its ability to leverage unlabeled industrial data to mitigate labeled data scarcity.

**Conflict of Interest Disclosure**

The author Josh Qixuan Sun conducted this work during an internship at Kisoji Biotech and received financial support from Kisoji Biotech. This project was also supported in part by Mitacs. The authors declare no other financial conflicts of interest that could reasonably be perceived to influence the work.

## 2. Related Work

**Preference learning for protein language models.** For general protein design, ProteinDPO (Xu et al., 2025) integrates structural feedback from folding simulators (e.g., AlphaFold) to optimize inverse folding models, significantly improving TM-scores on CATH benchmarks. ResiDPO (Xue et al., 2025) further refines this by introducing residue-level preference signals based on local confidence scores (pLDDT), enhancing the success rate of complex enzyme design. In the domain of antibody engineering, AbDPO (Zhou et al., 2024) and AlignAb (Wen et al., 2024) utilize physics-based energy functions to align diffusion models toward Pareto-optimal binders, balancing binding affinity with structural stability. To address the conservatism of KL-

regularized DPO, SimBinder-IF (Zhao et al., 2025) employs SimPO, a reference-free objective with a calibrated margin, achieving state-of-the-art Spearman correlation in zero-shot antibody-antigen affinity prediction. Beyond proteins, the paradigm has extended to RNA design through RiboPO (Sun et al., 2025), which aligns sequences with thermodynamic stability and geometric fidelity. To address the computational bottlenecks, g-DPO (Ferragu et al., 2025) implements a scalable framework that prunes redundant sequence pairs, achieving significant training acceleration across protein fitness and stability tasks. Collectively, these works demonstrate that preference-based fine-tuning effectively bridges the gap between evolutionary probability distributions and specific biophysical engineering goals.

**Protein developability prediction.** Computational approaches for developability have evolved from heuristic-based servers like Protein-Sol (Hebditch et al., 2017) to sophisticated deep learning architectures. Geometric approaches, such as GVP-GNN (Kim et al., 2023) and D-GNN (Khade et al., 2023), utilize equivariant graph neural networks to model the 3D spatial relationships and surface patches of residues, which are critical for predicting thermostability ($T_m$) and solubility. However, the current state-of-the-art is increasingly dominated by finetuning pLMs. Large-scale models such as ProtBERT (Brandes et al., 2022) leverage masked language modeling on hundreds of millions of sequences to capture structural and functional constraints without explicit supervision. Recent research highlights the efficacy of task-specific adaptation: ESM-1v (Meier et al., 2021) demonstrates high accuracy in predicting antibody non-specificity through logistic regression on its embeddings. NetSolP (Thumuluri et al., 2022) finetunes embeddings using a Transformer architecture to predict protein solubility and purification suitability. DeepSTABp (Jung et al., 2023) integrates PLM representations with experimental metadata to regress continuous $T_m$ values. RP3Net (Tankhilevich et al., 2026) takes the presentation from protein language model and trains a linear layer to predict whether the given antibody sequence is expressible or not. Overall, most existing pLM-based developability predictors follow a paradigm, where a pretrained pLM is frozen and its final-layer embeddings are used as features for downstream regression or classification heads. While effective, this strategy does not explicitly modify the sequence-level likelihood of the pLM. In contrast, preference learning methods, as discussed above, directly finetune the generative model itself using pairwise or ranked supervision, enabling the model to align its sequence probabilities with developability-related objectives.

# 3. Method

## 3.1. Problem Setup

We consider antibody expression ranking as a sequence-level scoring problem. An antibody sequence is represented as $y = (s_1, s_2, \ldots, s_L)$, where each amino acid token $s_i$ belongs to the standard set of 20 residues. We focus on variable-length VHH antibody sequences and make no assumption of fixed length.

Given a sequence $y$, the objective is to assign a scalar score that reflects its expression behavior, such that non-expressible sequences are ranked lower than expressible ones, and among expressible antibodies, sequences with higher expression yields are ranked higher. This formulation naturally supports both ranking-based evaluation and binary expressibility prediction without requiring separate task-specific models.

## 3.2. Sequence Scoring via Pseudo-Log-Likelihood

We derive sequence-level scores from protein language models (pLMs) using pseudo-log-likelihood (PLL) (Salazar et al., 2020; Meier et al., 2021). For a sequence $y = (s_1, \ldots, s_L)$, the PLL score is defined as

$$\mathrm{PLL}(y) = \frac{1}{L} \sum_{i=1}^{L} \log p_\theta(s_i \mid y_{\backslash i}),$$

where $y_{\backslash i}$ denotes the sequence with the $i$-th position masked. We use the average PLL as a length-normalized scalar score for ranking antibody sequences.

This scoring function induces a total ordering over sequences and can be directly applied in a zero-shot manner. To obtain a binary expressibility prediction from PLL scores, we adopt a percentile-based thresholding scheme. Specifically, given a training set with a known fraction of expressible sequences, we compute PLL scores for all training samples and select the corresponding percentile as a global decision threshold. This threshold is then applied consistently to validation and test sets, yielding a binary classifier without introducing additional trainable parameters.

## 3.3. Preference Optimization with Protein Language Models

While pseudo-log-likelihood provides a meaningful sequence-level score, it does not directly incorporate available supervision. We therefore adopt a preference-based learning framework to adapt protein language models using heterogeneous supervision signals.

### 3.3.1. DIRECT PREFERENCE OPTIMIZATION

Preference learning formulates antibody expression ranking through pairwise comparisons, and we adapt the Direct Preference Optimization (DPO) (Rafailov et al., 2023) framework. DPO optimizes a policy model $\pi_\theta$ to better satisfy preferences compared to a reference model $\pi_{\text{ref}}$ (typically the initial pretrained model), using a dataset $\mathcal{D}$ of preference pairs ($y_w, y_l$) for a given prompt $x$, where $y_w$ is preferred over $y_l$. The DPO objective maximizes the likelihood of preferred responses while regularizing against large deviations from the reference model via a KL divergence penalty:

$$\mathcal{L}_{\text{DPO}}(\pi_\theta; \pi_{\text{ref}}) = -\mathbb{E}_{(x, y_w, y_l) \sim \mathcal{D}} \Big[$$
$$\log \sigma \Big( \beta \log \frac{\pi_\theta(y_w \mid x)}{\pi_{\text{ref}}(y_w \mid x)} - \beta \log \frac{\pi_\theta(y_l \mid x)}{\pi_{\text{ref}}(y_l \mid x)} \Big) \Big],$$

where $\sigma$ is the sigmoid function, and $\beta$ is a hyperparameter controlling the strength of the preference relative to the regularization. In our context, $x$ is none, $y_w$ and $y_l$ are candidate antibody sequences, $\pi_{ref}$ is a pretrained pLM, and $\pi_\theta$ is the model being finetuned. This formulation allows both strong and weak supervision to be expressed uniformly as relative preferences, without introducing explicit reward models or task-specific heads.

### 3.3.2. LIKELIHOOD APPROXIMATION FOR PROTEIN LANGUAGE MODELS

The DPO objective relies on evaluating log-probabilities of sequences under both the policy model $\pi_\theta$ and the reference model $\pi_{\text{ref}}$. However, protein language models are typically trained with masked language modeling objectives and do not define tractable autoregressive likelihoods of the form $\log \pi(y \mid x)$. To address this, we calculate sequence log-likelihood induced by the policy $\pi$ using pseudo-log-likelihood (PLL) (Salazar et al., 2020; Meier et al., 2021):

$$\log \pi(y \mid x) = \text{PLL}_\pi(y) = \frac{1}{L} \sum_{i=1}^{L} \log \pi(s_i \mid y_{\setminus i}),$$

where $y_{\setminus i}$ denotes the sequence with the $i$-th position masked.

Under this calculation, the log-probability ratios in the DPO objective are replaced by differences in PLL scores. However, computing $\text{PLL}_\pi(y)$ exactly requires one forward pass per sequence position, which is computationally expensive during preference optimization. Crucially, DPO only depends on relative scores between preferred and less preferred sequences. For a preference pair $(y_w, y_l)$, we define the union mask:

$$\mathcal{M}(y_w, y_l) = \{ i \mid (y_w)_i \neq (y_l)_i \}.$$

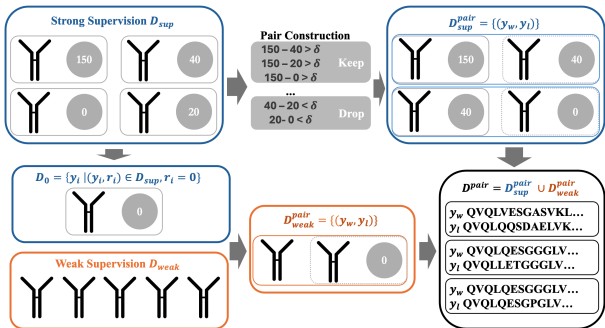

*Figure 1.* Construction of preference pairs. Strong supervision (top) uses yield differences with margin $\delta$; weak supervision (bottom) pairs immunization-derived sequences with zero-yield antibodies. Both contribute to the unified preference dataset $\mathcal{D}_{\text{pair}}$.

To make the definition of $\mathcal{M}(y_w, y_l)$ well-defined for antibody sequences with variable lengths, we first align sequences using a standardized numbering scheme. Specifically, we map each antibody sequence to IMGT numbering, which assigns homologous positions across variable-length variable regions. All sequence comparisons and masking operations are then performed in the aligned IMGT coordinate space.

We perform a single forward pass on the partially masked sequence. The log-score of $y_w$ under policy $\pi$ is then approximated as (Zhao et al., 2024; Hawkins-Hooker et al.; Ferragu et al., 2025):

$$\text{PLL}_\pi(y_w) \approx \frac{1}{|\mathcal{M}|} \sum_{i \in \mathcal{M}} \log \pi \big( (y_w)_i \mid y_{\setminus \mathcal{M}} \big),$$

with an analogous expression for $y_l$. Positions outside $\mathcal{M}$ need not be evaluated, as their contributions cancel in the score difference used by DPO.

### 3.3.3. PREFERENCE PAIR CONSTRUCTION

We construct preference pairs from two supervision sources through a unified pipeline, illustrated in Fig. 1. Both strong and weak supervision are converted into pairwise comparisons and merged into a single preference dataset.

**Strong supervision.** Let $\mathcal{D}_{\text{sup}} = \{(y_i, r_i)\}_{i=1}^{N}$ denote the set of antibody sequences with measured expression yields $r_i \in \mathbb{R}_{\geq 0}$. From this set, we construct preference pairs by comparing yield values. Specifically, for any pair $(y_i, y_j)$, we define a preference whenever $r_i - r_j \geq \delta$ where $\delta > 0$ is a predefined yield margin. In this case, $y_i$ is treated as the preferred sequence $y_w$ and $y_j$ as the less preferred sequence $y_l$. Pairs that do not satisfy this margin are discarded to reduce ambiguity due to experimental noise. In this work, we set $\delta = 30$.

**Weak supervision.** Weak supervision is provided by a large collection of immunization-derived antibody sequences, denoted as $\mathcal{D}_{\text{weak}} = \{y_k\}_{k=1}^M$. These sequences lack quantitative expression measurements and are treated as weak positive supervision.

To induce preferences without assigning explicit negative labels, we pair weakly supervised sequences against strongly supervised sequences with zero measured expression yield. Let $\mathcal{D}_0 = \{\, y_i \mid (y_i, r_i) \in \mathcal{D}_{\text{sup}}, \ r_i = 0 \,\}$ denote the subset of non-expressible sequences. For each $y_k \in \mathcal{D}_{\text{weak}}$ and a sampled $y_0 \in \mathcal{D}_0$, we construct a preference pair $(y_w, y_l) = (y_k, y_0)$.

**Unified preference dataset.** The final preference dataset is defined as $\mathcal{D}_{\text{pair}} = \mathcal{D}_{\text{sup}}^{\text{pair}} \cup \mathcal{D}_{\text{weak}}^{\text{pair}}$, where $\mathcal{D}_{\text{sup}}^{\text{pair}}$ and $\mathcal{D}_{\text{weak}}^{\text{pair}}$ denote preference pairs constructed from strong and weak supervision, respectively. Both supervision sources are thus integrated into a single preference learning framework. In practice, the number of pairs derived from these two sources is highly imbalanced, with weak supervision pairs vastly outnumbering the supervised ones. To ensure that the model effectively learns from the scarce but high-quality quantitative data, we implement a sampling rate hyperparameter $\alpha$ to oversample $\mathcal{D}_{\text{sup}}^{\text{pair}}$ relative to $\mathcal{D}_{\text{weak}}^{\text{pair}}$. This ensures a balanced gradient signal from both heterogeneous sources throughout the optimization process. The impact of different sampling rates will be further discussed in our ablation study.

### 3.4. Training Procedure

Training proceeds in two stages: domain-aligned continual pretraining with masked language modeling, followed by preference optimization.

**Stage I: Continual pretraining.** We first perform continual pretraining of the pretrained protein language model using a standard masked language modeling objective on the weakly supervised dataset $\mathcal{D}_{\text{weak}}$. This stage adapts the model to the antibody-specific sequence distribution and is sometimes referred to as *evo-tuning* (Alley et al., 2019; Biswas et al., 2021; Hsu et al., 2022; Gordon et al.). No expression-related supervision is used at this stage.

**Stage II: Preference finetuning.** Starting from the domain-aligned initialization, we apply DPO using the preference dataset $\mathcal{D}_{\text{pair}}$. Model parameters are optimized to satisfy relative preferences between antibody sequences under the DPO objective, integrating both strong quantitative supervision and weak positive supervision.

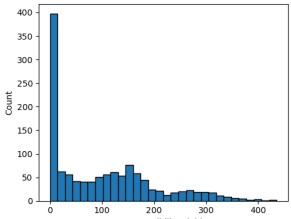
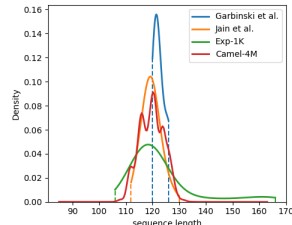

*Figure 2.* Histogram of Exp-1K yield score.

*Figure 3.* Sequence length density across different datasets.

## 4. Experiments

### 4.1. Datasets

To evaluate our preference-based learning framework under realistic antibody discovery conditions, we curate two internal datasets: **Exp-1K** for supervised evaluation and **Camel-4M** for large-scale weak supervision. The statistical comparison with existing public benchmarks is summarized in Table 1.

**Exp-1K: Labeled Expression Benchmark.** This dataset comprises 1254 unique VHH sequences derived from internal antibody designs. Unlike public datasets that primarily consist of expressible sequences, Exp-1K explicitly includes approximately 27% (335 sequences) of non-expressible candidates (no detectable expression band, expressibility Yield (mg/L) $<$ 1 mg/L). As illustrated in Figure 2, the yield distribution is highly skewed with a distinct zero-yield peak. This sparsity and imbalance make Exp-1K a more challenging and realistic benchmark than existing small-scale datasets.

**Camel-4M: Large-scale Weak Supervision.** To provide a broad representation of the antibody sequence space, we utilize Camel-4M, a corpus of 4.2 million non-redundant VHH sequences. These sequences were derived from PBMC samples of immunized camelid species, including alpacas and llamas. Figure 3 shows the sequence length density across different datasets. We treat the sequences from the immune maturation process as weak positive supervision.

### 4.2. Experimental Setup

**Model Backbones and Initialization**. We evaluate our framework across three representative backbones: AntiBERTa2, IgBERT, and ProtBERT. All models utilize their default architectures without structural modifications. For our DPO-based approach, the reference model $\pi_{\text{ref}}$ is initialized from the weights after Stage I (continual pretraining) rather than the vanilla pLMs, ensuring a consistent starting point for preference alignment.

**Training Details**. Our training pipeline consists of two stages. Stage I: Continual Pretraining (CPT). We perform CPT on the Camel-4M dataset using the standard masked

| Backbone | Type | Method | Stage I | Binary | | Ranking | | | Composite |
|---|---|---|---|---|---|---|---|---|---|
| | | | | AUC ↑ | MCC ↑ | Tau ↑ | SCC ↑ | Recall ↑ | Avg ↑ |
| AntiBERTa2 (Barton et al., 2024) | aLM | Zero-shot | × | 52.6 | 6.8 | -0.2 | -4.1 | 11.5 | - |
| | | Zero-shot | ✓ | 51.4 | 13.2 | -0.1 | -0.1 | 0.1 | - |
| | | Supervised | × | 74.5 | 29.1 | 37.6 | 42.2 | 42.3 | 33.1 |
| | | Supervised | ✓ | 76.2 | 21.8 | 38.4 | 42.4 | 42.3 | 29.0 |
| | | Ours (Stage II) | × | 77.2 | 26.7 | 33.8 | 32.8 | 38.5 | 30.8 |
| | | Ours (Stage I+II) | ✓ | **82.4** | **43.7** | **46.3** | **50.8** | **46.2** | **45.0** |
| IgBERT (Kenlay et al., 2024) | aLM | Zero-shot | × | 50.4 | -12.6 | -0.2 | -0.9 | 15.4 | - |
| | | Zero-shot | ✓ | 56.0 | -1.5 | 9.0 | 9.7 | 27.0 | - |
| | | Supervised | × | **78.6** | 7.5 | 40.0 | 43.5 | 34.6 | 17.2 |
| | | Supervised | ✓ | 75.7 | 36.8 | 38.3 | 45.8 | 46.2 | 37.5 |
| | | Ours (Stage II) | × | 73.0 | 17.1 | 34.3 | 41.5 | 46.2 | 24.2 |
| | | Ours (Stage I+II) | ✓ | 74.8 | **37.5** | **42.9** | **54.7** | **61.5** | **40.1** |
| ProtBERT (Brandes et al., 2022) | pLM | Zero-shot | × | 43.6 | -3.7 | -8.8 | -11.0 | 11.5 | - |
| | | Zero-shot | ✓ | 54.1 | 17.4 | 4.0 | 2.8 | 26.9 | 8.4 |
| | | Supervised | × | 77.8 | 16.7 | 37.3 | 40.4 | 34.6 | 24.9 |
| | | Supervised | ✓ | 76.4 | 26.1 | 37.1 | 42.1 | 38.5 | 31.1 |
| | | Ours (Stage II) | × | 74.5 | 28.6 | 29.8 | 27.3 | 19.2 | 29.2 |
| | | Ours (Stage I+II) | ✓ | **78.0** | **40.1** | **40.5** | **44.5** | **53.8** | **40.3** |

*Table 2.* Main results on the Exp-1K test set across two antibody language model (aLM) and one protein language model (pLM) backbone. AUC, MCC, and Recall evaluate binary expressibility prediction, while Kendall's Tau (Tau) and Spearman's correlation coefficient (SCC) evaluate yield ranking. Avg denotes the geometric mean of MCC and Tau. Recall is reported at top 20% according to the predicted sequence scores.

language modeling (MLM) objective with a 15% mask ratio. Training is conducted for 1 epoch with a learning rate of $1 \times 10^{-5}$, a batch size of 128, and a maximum sequence length of 168. Stage II: Preference Fine-tuning. We use the AdamW optimizer with a learning rate of $1 \times 10^{-4}$, 1K warmup steps, and a weight decay of 0.01. The KL-divergence regularization parameter $\beta$ is set to 0.1, and the yield margin $\delta$ for constructing preference pairs is fixed at 30. For the main results, we use a sampling rate $\alpha = 1000$ to balance the weak and strong supervision.

**Baselines and Fair Comparison**. To demonstrate the efficacy of preference-based modeling, we implement a standard supervised baseline optimized with Mean Squared Error (MSE) loss. It utilizes the same pLM backbones followed by an attention pooling layer and a linear head. For a rigorous comparison, we report results using vanilla pLM weights and Stage I-tuned weights.

**Evaluation Metrics**. Following established benchmarks such as ProteinGym and FLAb, we evaluate model performance across two dimensions: sequence ranking and binary classification. (1) Ranking Metrics: To evaluate the correlation between predicted scores and experimental yields, we report Kendall's $\tau$ and Spearman's Correlation Coefficient (SCC). While SCC is a standard metric in protein fitness landscapes, it is sensitive to "ties" (identical ranks). Given that approximately 27% of the Exp-1K dataset consists of non-expressible sequences (yield = 0), the resulting heavy ties in ground-truth labels make SCC less robust. There-

fore, we adopt Kendall's $\tau$ as our primary ranking metric due to its superior handling of tied observations. To provide a more granular view, SCC is calculated specifically on the subset of expressible sequences (yield > 0). (2) Binary Metrics: We use the Matthews Correlation Coefficient (MCC) to assess the model's ability to distinguish between expressible and non-expressible antibodies. (3) Composite Metric (Avg): We introduce a unified score calculated as the geometric mean of MCC and Kendall's $\tau$. We prioritize the geometric mean over the arithmetic mean to penalize models that achieve high performance in one dimension at the significant expense of the other, thereby ensuring a balanced optimization for real-world antibody screening.

**Data Split and Implementation**. The Exp-1K dataset is partitioned into training, validation, and test sets with an 80/10/10 ratio. Model selection is performed based on validation performance, and final results are reported on the unseen test set. Experiments are conducted on NVIDIA H100 GPUs using bf16 precision, with batch sizes adjusted according to the backbone scale.

### 4.3. Main Results

Table 2 reports the main results on the Exp-1K test set across two antibody language model backbones (aLM) and one protein language model (pLM) backbone. We evaluate both binary expressibility prediction and yield ranking to reflect the dual nature of the task. Across all evaluated metrics, our two-stage method consistently achieves the

strongest overall performance. For all three backbones, the proposed approach outperforms baselines mostly across all metrics. This demonstrates that preference optimization jointly improves ranking and classification performance.

**Limitations of zero-shot baselines.** We first observe that zero-shot baselines perform poorly across all metrics, regardless of whether continual pretraining is applied. In particular, Kendall's Tau and MCC are close to zero or even negative. This suggests that sequence likelihood under a pretrained language model is weak for expressibility. This behavior is expected, as expressibility differs fundamentally from properties such as thermostability or evolutionary fitness. Many antibody sequences in the evaluation set are synthetically designed and do not correspond to naturally occurring sequences. As a result, their probability under the natural sequence distribution modeled by a protein language model is not strongly correlated with downstream expression outcomes. Continual pretraining alone does not resolve this mismatch, since it optimizes the same likelihood-based objective.

**Limited gains from Stage I with supervised heads.** Next, we compare supervised learning baselines with and without Stage I continual pretraining. Applying continual pretraining yields marginal or inconsistent improvements, and in some cases leads to performance degradation. This trend is visible across multiple metrics. This result indicates that representations optimized via continual pretraining are not necessarily aligned with the features required by downstream supervised objectives. Simply initializing a supervised head from a continually pretrained backbone does not guarantee better expressibility prediction, especially when labeled data remain limited.

**Preference learning vs. Stage I + supervised learning.** A key comparison in Table 2 is between Stage I CPT followed by supervised learning and our preference-based post-training. This comparison directly addresses the question of whether continual pretraining combined with standard regression is sufficient for expressibility prediction. Despite using the same continually pretrained backbone, our method substantially outperforms supervised baselines across all reported metrics. In particular, gains in Kendall's Tau and MCC indicate improved alignment with both ranking and classification objectives. This highlights the importance of preference-based optimization, which directly matches the relative supervision signal available from weakly labeled data and avoids objective mismatch.

**Synergy between Stage I and Stage II.** Comparing "Ours (Stage II)" with "Ours (Stage I+II)" reveals that Stage I continual pretraining is essential for maximizing performance. For AntiBERTa2, the full pipeline boosts MCC from 26.7 to 43.7 and Tau from 33.8 to 46.3. This demonstrates that Stage I creates a foundational representation space that is

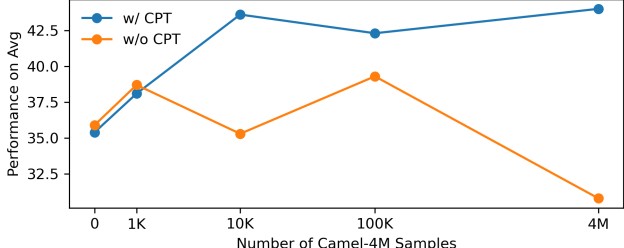

*Figure 4.* Scaling with weak supervision. Results are shown with and without Stage I continual pretraining (CPT), while keeping a consistent exposure ratio between supervised and weak pairs.

uniquely synergistic with Stage II preference optimization, significantly outperforming the use of Stage II alone.

### 4.4. Scaling with Weak Supervision

To investigate how our framework benefits from increasing scales of weak supervision, we evaluate the model performance by varying the size of $\mathcal{D}_{\text{weak}}$ from 1K to 4.2M. We conduct this analysis with AntiBERTa2 backbone under two settings: (i) w/ Stage I continual pretraining (CPT) on Camel-4M, and (ii) w/o Stage I, where the preference learning starts directly from the vanilla pLM. To ensure a fair comparison, the sampling rate $\alpha$ is dynamically adjusted to maintain a consistent exposure ratio between supervised pairs and weak pairs.

As shown in Figure 4, the average metric (Avg) generally increases with the amount of weak supervision for both backbones. For the w/o Stage I setting, the curve exhibits noticeable fluctuations as the data scale grows. In contrast, when Stage I pretraining is applied, the improvement with additional weak supervision is more stable and sustained, indicating that the proposed preference learning setup can continue to benefit from larger amounts of unlabeled immunization data.

### 4.5. Sensitivity to Data Composition

The balance between high-fidelity supervised pairs $\mathcal{D}_{\text{sup}}$ and large-scale weak preference pairs $\mathcal{D}_{\text{weak}}$ is crucial for effective preference learning. To manage the vast size disparity between these sets, we introduce a sampling hyperparameter $\alpha$ to modulate the exposure frequency of supervised data. Formally, a weak preference pair is selected with probability $p_{\text{weak}} = N_{\text{weak}}/(N_{\text{weak}} + \alpha \cdot N_{\text{sup}})$, where $N_{\text{weak}}$ and $N_{\text{sup}}$ denote the total number of available pairs in each set respectively. Detailed calculations are in Appendix A.5.

As shown in Table 3, we evaluate the model performance across different $\alpha$ values. Contrary to the observation in standard multi-task learning, the performance exhibits a clear sensitivity to the sampling ratio. We find that a default value of $\alpha = 1000$ (corresponding to $p_{\text{weak}} \approx 74\%$) provides the optimal trade-off. Lower values of $\alpha$ lead to

| $\alpha$ | $p_{weak}$ | MCC | Tau | SCC |
|------|------|------|------|------|
| 10 | 99.7% | 13.9 | 12.2 | 8.7 |
| 100 | 96.7% | 21.7 | 14.2 | 0.0 |
| 1000 | 74.6% | 40.1 | 40.5 | 44.5 |
| 5000 | 37.1% | 46.9 | 38.0 | 37.4 |
| 10000 | 22.7% | 31.1 | 30.7 | 26.8 |

*Table 3.* Sensitivity analysis of the sampling hyperparameter $\alpha$. $p_{weak}$ represents the theoretical probability of sampling a weak preference pair in each training step.

| Method | $\Delta$ Avg |
|------|------|
| Ours (Full Framework) | 5.5 |
| w/o Weak Data | 14.1 |
| CPT + Supervised Regression | 16.3 |

*Table 4.* Generalization gap $\Delta = |\text{Avg}_{val} - \text{Avg}_{test}|$ across different settings. Lower $\Delta$ indicates better generalization.

the "swamping" of the high-quality supervised signal by the massive weak data, while excessively high values cause the model to overfit the small-scale Exp-1K dataset, sacrificing the general structural priors offered by the immunization data.

### 4.6. Regularization and Stability

A fundamental question arises: does weak preference learning act as a regularizer? We investigate this by examining the optimization dynamics and generalization gaps across different learning paradigms.

**Generalization Gap.** As summarized in Table 4, we report the discrepancy $\Delta = |\text{Avg}_{val} - \text{Avg}_{test}|$ for the average metric. Our full framework exhibits the tightest gap, whereas removing weak data or switching to a standard supervised regression objective significantly widens this gap. This suggests that the large-scale $\mathcal{D}_{weak}$ anchors the model within a biologically viable sequence space, preventing it from fitting experimental noise specific to the 1K samples.

**Optimization Stability.** Figure 5 illustrates the validation performance trajectories over training. To enable a fair comparison between the step-based DPO (evaluated every 250 steps) and the epoch-based regression (evaluated every epoch), we align their training progress on a relative scale. We observe that models trained without weak supervision exhibit erratic performance oscillations and "spiky" validation curves. In contrast, our unified preference objective yields a remarkably smooth convergence trajectory, suggesting that the massive preference pairs provide a flatter and more robust optimization landscape.

### 4.7. Analysis of Preference Construction

To investigate the intrinsic behavior of preference-based learning in antibody expression modeling, we first evaluate whether DPO remains effective under a strictly supervised

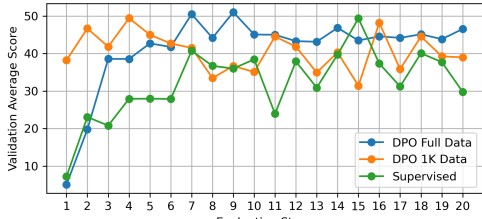

*Figure 5.* Validation performance trajectories over training for different methods.

| Method | MCC $\uparrow$ | Tau $\uparrow$ | SCC $\uparrow$ |
|------|------|------|------|
| Supervised | 29.1 | 37.6 | 42.2 |
| w/o Camel-4M | 32.8 | 35.1 | 36.2 |
| Ours | 43.7 | 46.3 | 50.8 |

*Table 5.* Comparison between supervised regression and DPO on the Exp-1K labeled set (w/o Camel-4M).

setting. Table 5 compares the performance of the supervised baseline against DPO trained solely on the 1.2k labeled sequences from Exp-1K. Interestingly, we observe that without the inclusion of large-scale weak supervision (Camel-4M), the pure DPO approach yields slightly inferior results compared to direct regression. We attribute this to the fact that preference learning provides a sparser gradient signal than point-wise regression when the labeled dataset is extremely small. This strongly emphasizes the necessity of incorporating massive weakly labeled data within the preference learning framework.

Building on this, we examine the impact of sequence similarity on preference construction. We selected two specific thresholds to partition the dataset, which correspond to the natural local minima (Figure 6) in the pairwise similarity distribution of Exp-1K. This allows us to compare the effectiveness of learning from closely related mutants versus more divergent antibody scaffolds. Our results (Table 6) demonstrate that utilizing the full distribution consistently yields superior performance. This justifies our choice to utilize the entire sequence space without heuristic filtering.

## 5. Conclusion

In this work, we present a unified preference-based framework for antibody expression modeling, addressing the scarcity of labeled experimental data by integrating large-scale weak supervision. By adapting Direct Preference Optimization (DPO) to protein language models with a union-masked log-likelihood objective, we successfully leverage massive immunization data to learn biological priors. Our experiments demonstrate that while traditional binary classifiers remain highly effective for simple expressibility categorization, our preference-based approach offers superior performance in complex ranking tasks and exhibits significantly better generalization and optimization stability. This

| Threshold | MCC ↑ | Tau ↑ | SCC ↑ |
|---|---|---|---|
| No threshold | 43.7 | 46.3 | 50.8 |
| 0.31 | 35.7 | 37.6 | 38.5 |
| 0.59 | 6.4 | 18.3 | 19.4 |

*Table 6.* Impact of similarity-based pair filtering on DPO performance. Thresholds are chosen based on the local minima of the dataset's similarity distribution.

discovery suggests that preference learning acts as a robust biophysical regularizer, providing a scalable solution for fine-grained antibody developability prediction. Future research will focus on hybrid architectures that combine the discriminative power of binary supervision with the ranking precision of preference-based alignment.

## Impact Statement

This paper presents work whose goal is to advance the field of AI for drug discovery. There are many potential societal consequences of our work, none of which we feel must be specifically highlighted here.

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

# A. Experimental Implementation Details

## A.1. Dataset Statistics and Preprocessing

**Exp-1K Statistics.** The Exp-1K dataset serves as our primary benchmark for quantitative yield prediction. The sequence counts for the 80/10/10 split are 1,003 (Train), 125 (Val), and 126 (Test).

**IMGT Alignment.** We use ANARCI (Dunbar & Deane, 2016) to number all sequences according to the IMGT standard numbering (Lefranc et al., 2003a;b). Sequences are mapped to 128 standard positions, with missing residues filled by gap tokens $(-)$. This alignment is crucial for the position-wise log-likelihood calculation in our DPO framework.

**Hamming Similarity.** To quantify the similarity between two antibody sequences, we adapt the Hamming similarity based on the IMGT numbering scheme. Given two antibody sequences $s_1$ and $s_2$, we first map them to a standardized coordinate system of IMGT residue indices $\mathcal{I}$ (e.g., positions 1 to 128). Let $\text{pos}(s, i)$ be the amino acid residue of sequence $s$ at the IMGT index $i \in \mathcal{I}$. The similarity score $h(s_1, s_2)$ is defined as the fraction of identical residues across the intersection of their aligned positions:

$$h(s_1, s_2) = \frac{\sum_{i \in \mathcal{I}_{s_1} \cap \mathcal{I}_{s_2}} \mathbb{I}(\text{pos}(s_1, i) = \text{pos}(s_2, i))}{|\mathcal{I}_{s_1} \cap \mathcal{I}_{s_2}|},$$

where $\mathbb{I}(\cdot)$ is the indicator function and $|\mathcal{I}_{s_1} \cap \mathcal{I}_{s_2}|$ represents the number of shared IMGT positions.

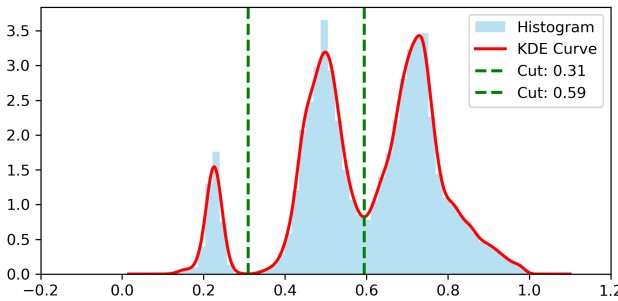

*Figure 6.* Pairwise similarity distribution of the Exp-1K dataset. The similarity is calculated as the IMGT-aligned Hamming identity across all sequence pairs.

**Pairwise Similarity Distribution.** To facilitate the analysis of preference construction (Section 4.7), we compute the pairwise Hamming similarity for all sequences in Exp-1K based on the IMGT-aligned positions. As illustrated in Figure 6, the similarity distribution exhibits a distinct tri-modal pattern, characterizing different levels of sequence divergence (e.g., clonal variants vs. divergent scaffolds). To rigorously test the effect of similarity-based filtering, we select the two local minima (valleys) of this distribution, $d_1 = 0.31$ and $d_2 = 0.59$, as the thresholds for the sensitivity analysis in Table 6. This systematic approach ensures that our thresholds are grounded in the intrinsic structural hierarchy of the antibody library.

## A.2. Model Configurations

Table 7 summarizes the architectural details of the two aLMs and one pLM used as backbones.

| Model | Layers | Heads | Dim | Params | HuggingFace Path |
|---|---|---|---|---|---|
| AntiBERTa2 | 16 | 16 | 1024 | 202M | alchemab/antiberta2 |
| IgBERT | 30 | 16 | 1024 | 420M | Exscientia/IgBert |
| ProtBERT | 30 | 16 | 1024 | 420M | Rostlab/prot_bert |

*Table 7.* Architectural configurations of the pLM backbones.

## A.3. Hyperparameter Details

We use the AdamW optimizer with default $\beta_1 = 0.9, \beta_2 = 0.999$. No gradient clipping is applied. Specific batch sizes and GPU memory usage are listed in Table 8.

| Model | Batch Size | Peak Memory | Precision |
|---|---|---|---|
| *Stage I* | | | |
| AntiBERTa2 | 256 | 32.0GB | bf16 |
| IgBERT | 256 | 45.5GB | bf16 |
| ProtBERT | 256 | 45.5GB | bf16 |
| *Stage II* | | | |
| AntiBERTa2 | 128 | 32.1GB | bf16 |
| IgBERT | 128 | 45.7GB | bf16 |
| ProtBERT | 128 | 45.7GB | bf16 |

*Table 8.* Computational details and batch sizes per GPU (H100).

## A.4. Convergence and Early Stopping Criteria

We observed that different training stages require distinct convergence strategies to maintain a balance between domain adaptation and the risk of overfitting to the relatively scarce labeled data.

**Stage I (Continual Pre-training):** For the domain-alignment phase, the model is trained for a fixed duration of 1 epoch on the Camel-4M dataset. Given the large-scale nature of the unlabeled data, a single pass is sufficient to adapt the model to the antibody-specific sequence distribution without collapsing the general protein representations learned during initial pre-training.

**Stage II (Preference Fine-tuning):** n the preference optimization phase, we employ an Early Stopping mechanism to prevent over-optimization on preference pairs. Training is terminated if the validation metric (specifically, the geometric mean of Matthews Correlation Coefficient and Kendall's $\tau$) fails to improve for 10 consecutive evaluation steps. (1) For Supervised Baselines: Evaluation occurs once per epoch. Thus, the patience for early stopping is equivalent to 10 epochs. (2) For DPO Fine-tuning: Due to the higher density of preference pairs, evaluation occurs every 250 gradient steps. Consequently, the early stopping patience is equivalent to 2500 training steps.

## A.5. Probabilistic Sampling for Supervision Balancing

A key challenge in Stage II is the extreme imbalance between the number of strong supervision pairs $|\mathcal{D}_{\text{sup}}^{\text{pair}}|$ and weak supervision pairs $|\mathcal{D}_{\text{weak}}^{\text{pair}}|$. To prevent the weak supervision signal from overwhelming the high-quality quantitative data, we implement a probabilistic sampling strategy at the `__getitem__` level of our data loader.

We define a sampling rate hyperparameter $\alpha$ (set to 1000 in our main experiments) to oversample the supervised pairs. The probability of selecting a sample from either the strong supervision set ($S$) or the weak supervision set ($L$) is calculated as follows:Let $N_S = |\mathcal{D}_{\text{sup}}^{\text{pair}}|$ and $N_L = |\mathcal{D}_{\text{weak}}^{\text{pair}}|$. The selection probabilities $P_S$ and $P_L$ are defined as:

$$P_S = \frac{N_S \cdot \alpha}{(N_S \cdot \alpha) + N_L}, \quad P_L = \frac{N_L}{(N_S \cdot \alpha) + N_L}$$

During training, for each data request, a random variable $u \sim \text{Uniform}(0, 1)$ is generated. If $u < P_S$, a pair is sampled from the supervised set; otherwise, it is sampled from the weakly supervised set. This ensures that the expected ratio of supervised gradient updates remains controllable and consistent regardless of the raw size of the unlabeled corpus.

## A.6. Mathematical Definitions of Metrics

**Matthews Correlation Coefficient (MCC).** Unlike ROC-AUC, which can be overly optimistic on imbalanced data, MCC provides a balanced score even if the classes are of very different sizes:

$$\text{MCC} = \frac{TP \times TN - FP \times FN}{\sqrt{(TP + FP)(TP + FN)(TN + FP)(TN + FN)}} \tag{1}$$

**Kendall's $\tau$ (Tau-b).** To handle the heavy ties (Yield=0) in Exp-1K, we use the standard $\tau_b$ which adjusts for ties:

$$\tau_b = \frac{C - D}{\sqrt{(C + D + T_x)(C + D + T_y)}}, \tag{2}$$

where $C$ is the count of concordant pairs, $D$ is the count of discordant pairs, $T_x$ is the number of ties only in $x$, and $T_y$ is the number of ties only in $y$.

**SCC on Expressible Subset.** We define the filtered Spearman correlation coefficient (SCC) as:

$$\text{SCC}_{\text{expr}} = \text{Spearman}(\{(y_i, \hat{y}_i) \mid y_i > 0\}) \tag{3}$$

**Composite Metric (Avg).** We use the geometric mean to penalize models that fail significantly in either classification or ranking:

$$\text{Avg} = \sqrt{\text{MCC} \times \tau_b} \tag{4}$$

## B. Binary Supervision with Weak Positive Labels

In this ablation, we examine whether large-scale weakly labeled data can be directly exploited by binary classification. We focus on the Supervised-Binary baseline using AntiBERTa2 as the backbone. Based on preliminary experiments, we observe that initializing from the original pretrained weights consistently outperforms continual pretraining for binary supervision, and therefore fix the backbone accordingly. We augment the labeled training set with subsets of Camel-4M, which are assumed to be expressible. To address the resulting class imbalance, we oversample the non-expressible class and report results under the best-performing sampling ratio.

| Camel-4M Size | AUC ↑ | MCC ↑ |
|---|---|---|
| None (Exp-1K only) | 86.2 | 35.7 |
| 1K | 85.3 | 41.6 |
| 10K | 85.9 | 51.9 |
| 100K | 87.8 | 56.7 |
| 1M | 86.7 | 50.7 |
| 4M | 87.1 | 48.3 |
| Ours (4M) | 82.3 | 43.7 |

*Table 9.* Effect of weak positive samples on supervised binary training using AntiBERTa2.

For the narrow task of binary expressibility classification, the standard supervised binary network outperforms our preference-based model (Table 9). We attribute this to the fact that a binary classifier directly optimizes the decision boundary for a single threshold, whereas our framework treats expressibility as a ranking problem within a much broader sequence space.

## C. Dataset Generation

### C.1. Exp-1K

All antibody sequences in Exp-1K were transiently transfected into mammalian expression systems (CHO or HEK293 cells). Antibody expression was evaluated using SDS-PAGE–based expression gels, and yield was estimated semi-quantitatively based on band intensity relative to internal standards. Reported yields are expressed in mg/L. Antibodies with no detectable expression band were assigned a yield of $< 1$ mg/L.

### C.2. Camel-4M

Camel-4M consists of antibody libraries derived from PBMC samples of immunized camelid species, including alpacas and llamas. Following immunization with diverse therapeutic targets, PBMCs were isolated and used to construct VHH antibody libraries, which were subsequently sequenced using next-generation sequencing (MiSeq) to obtain non-redundant VHH sequences. Because these libraries are derived from PBMCs of immunized animals, the resulting sequences predominantly represent mature, antigen-experienced B cells. Antibodies produced by such cells have undergone immune selection for proper folding and secretion, resulting in an enrichment of expressible VHHs.

### C.3. Evidence of why Camelid-derived sequences are mostly expressible.

**Theoretical evidence.** Camelid-derived VHHs are well known to exhibit high expression and solubility (Flajnik et al., 2011; Liang et al., 2015). Camelid species have uniquely evolved heavy-chain–only antibodies (HCAbs), in which the

antigen-binding domain (VHH) functions naturally without a paired light chain. As part of this evolutionary adaptation, VHHs contain canonical framework mutations, particularly in the former VH–VL interface within FR2, where hydrophobic residues are replaced by more hydrophilic ones to promote proper folding, solubility, and secretion. In contrast, conventional VH domains derived from VH–VL antibodies lack these canonical FR2 mutations and are evolutionarily optimized to pair with a light chain, making isolated VH-derived formats, including VH-based heavy-chain antibodies, generally more difficult to express (Zhang et al., 2018). Together, these evolutionary and structural features explain the enrichment of expressible sequences in camelid-derived VHH libraries.

**Internal evidence.** In addition to prior biological intuition, we provide direct internal experimental evidence supporting the assumption that camelid-derived antibody sequences are predominantly expressible. We randomly selected 17 VHH sequences from the camelid-derived pool and measured their recombinant expression yields under the same experimental protocol used for the Exp-1K dataset.

Among these 17 sequences, 16 showed detectable expression (yield $> 0$ mg/L), corresponding to an expressibility rate of $94.1\%$. This observation is consistent with the hypothesis that immunization-derived camel VHH sequences are mostly expressible.

To further characterize the expression behavior of these sequences, Table 10 reports the distribution of measured yield scores across several intervals. Rather than concentrating near the detection limit, most sequences exhibit non-trivial expression levels, indicating that camelid-derived sequences are not only binary-expressible, but often moderately to highly expressible.

| Yield range (mg/L) | Number of sequences |
|---|---|
| $= 0$ | 1 |
| $(0, 50)$ | 1 |
| $[50, 100)$ | 2 |
| $[100, 200)$ | 6 |
| $> 200$ | 7 |
| Total | 17 |

*Table 10.* Yield score distribution of 17 randomly sampled camelid-derived VHH sequences.

