# OpenReview forum: "Preference-based Antibody Expression Ranking: Scaling with Large-scale Weak Supervision"
_ICML.cc/2026/Conference — ICML 2026 regular_

### Official Review · Reviewer_DdoU · 2026-03-10

**Soundness:** 3
**Presentation:** 3
**Significance:** 3
**Originality:** 3
**Overall Recommendation:** 4
**Confidence:** 3

**Summary:**

This work addresses an important area of work: modeling antibody expression ranking due to limited labeled experimental data. The authors propose a two stage framework based on domain adaptation of LLMs followed by DPO based training. They exploit large-scale
weak biological supervision together with limited labeled experimental data to improve predictive models towards a scalable solution for antibody expressibility optimization in data-constrained settings. Overall its a very interesting work.

**Compliance With Llm Reviewing Policy:**

Affirmed.

**Key Questions For Authors:**

Following are the questions

1. Is there a possibility of having sequence overlaps (leakage)  between the Camel-4M and Exp-1K?
2. How robust is the model to full length antibody (generalization)?

**Limitations:**

Minor Weaknesses: Novelty is leveraging weak supervision in data scarcity regime rather than methods.
Reproducibility of results might be limited to due unavailable public data.
Also there is no discussion on the properties of antibodies such as solubility. Related comment is the paper is framed as a ranking rather than generative model, but it would be interesting to see the results when combined with a generative model.

**Strengths And Weaknesses:**

This is an critical area of research high practical impact. Antibody developability prediction is a known bottleneck in therapeutic pipelines leading to scarce experimentally validated data. The highlight of the paper is the tapping into the immunization derived sequences as weak biological supervision leading to training data augmentation. This can be adopted in any domain in protein design where data is limited.
The experiments are justified in selecting ranking metrics and classification metrics and a composite metric. The ablation shows indeed with only DPO the performance is worse.

---

> ### Author Rebuttal · Authors · 2026-03-31
>
> > **Is there a possibility of having sequence overlaps (leakage) between the Camel-4M and Exp-1K? (Question 1)**
>
> We appreciate the reviewer’s attention to data integrity. We can confirm that there is no data leakage between the Camel-4M and the evaluation sets of Exp-1K. Our response is grounded in the following points:
> * Strict Partitioning: We conducted a rigorous identity-based filtering process during dataset preparation. All sequences used for evaluation (the validation and test splits of Exp-1K) were strictly excluded from the Camel-4M dataset and the training split of Exp-1K.
> * Clarification on the 17 Sequences (Appendix D.3): As mentioned in the Appendix, 17 sequences from the Camel-4M pool were indeed cross-referenced with Exp-1K. The purpose of this overlap was purely for biological hypothesis validation, to empirically support our assumption that camelid-derived sequences are predominantly expressible (94.1% expressibility observed).
>
> > **How robust is the model to full length antibody (generalization)? (Question 2)**
>
> Theoretically, the algorithm has potential to apply to non-VHH antibodies if we have available data to train the model on VH-VL sequences.
>
> > **Novelty is leveraging weak supervision in data scarcity regime rather than methods.**
>
> Our paper identifies a specific **flaw in treating expression as a standard supervised learning task, as existing approaches often oversimplify the complex biological constraints involved.**
>
> **Our core insight, developed through industrial collaboration, is that naturally occurring antibody molecules provide a latent resource of weak supervision** where expression levels are usually high, even without exact labels. Standard supervised methods fail to capture this. We demonstrate that Preference Optimization provides the necessary mathematical interface to unify these two disparate data sources.
>
> Critically, our empirical results (Table 5) show that **directly applying these techniques to supervised data only is insufficient and can even be outperformed by simple MLP baselines**. This proves that the novelty and performance gains lie not in the algorithms themselves, but in the novel interaction between weak immunization signals and preference learning to solve the data-scarcity bottleneck in antibody design.
>
> > **Reproducibility of results might be limited to due unavailable public data.**
>
> We plan to release the dataset upon completion of the patenting process.
>
> > **No discussion on the properties of antibodies such as solubility.**
>
> We emphasize that while our evaluation focuses on expression, the framework is fundamentally generalizable:
>
> * Methodological Generality: Our proposed framework, including the IMGT-based alignment and the union-masked log-likelihood approximation, is fundamentally a general-purpose solution for adapting masked protein language models to any sequence-level ranking task. It can be directly applied to other properties where pairwise is available.
> * The "Weak Supervision" Paradigm: A core contribution of this paper is demonstrating how large-scale weak positive supervision can mitigate the scarcity of labeled data. We identified antibody expression as the property to validate this paradigm because of the unique biological link between camel-derived immunization data and expressibility.
>
> > **Importance of expressibility research**
>
> We would like to emphasize that **antibody expressibility is a critical developability constraint with direct implications for development cost and program risk**. Poorly expressed candidates introduce significant liability, requiring iterative redesign, increasing CMC burden, and delaying timelines. Accordingly, developability assessment is now recognized as an essential component of early-stage antibody discovery [1].
>
> This challenge is further amplified by recent advances in AI-driven protein and antibody generation. While these models enable rapid exploration of sequence space, they do not guarantee that generated sequences satisfy key developability constraints such as expression and stability, and still require substantial experimental validation [2, 3].
>
> As a result, expression prediction is not a secondary task but a critical gating factor for translating computational designs into viable therapeutic candidates. Our work directly addresses this gap by focusing on expression-aware ranking under realistic data scarcity.
>
> ref:
>
> [1] Zhang, Weijie, et al. "Developability assessment at early-stage discovery to enable development of antibody-derived therapeutics." Antibody Therapeutics 6.1 (2023): 13-29.
>
> [2] Cho, Yehlin, et al. "Stable de novo protein design via joint conformational landscape and sequence optimization." Nature Communications (2025).
>
> [3] Bennett, Nathaniel R., et al. "Atomically accurate de novo design of antibodies with RFdiffusion." Nature 649.8095 (2026): 183-193.

---

> > ### Author Rebuttal · Reviewer_DdoU · 2026-03-31
> >
> > Thank you for addressing the concerns. I maintain my score.

---

### Official Review · Reviewer_BN3p · 2026-03-12

**Soundness:** 2
**Presentation:** 3
**Significance:** 3
**Originality:** 3
**Overall Recommendation:** 4
**Confidence:** 4

**Summary:**

This paper proposes a preference-based learning framework for antibody expression ranking that incorporates Direct Preference Optimization (DPO) with protein language models via a union-masked log-likelihood objective. The authors introduce two proprietary datasets, Exp-1K and Camel-4M, which significantly exceed existing public benchmarks in scale and diversity. Experiments demonstrate consistent improvements over baselines for the antibody expression ranking task.

**Compliance With Llm Reviewing Policy:**

Affirmed.

**Final Justification:**

My concerns have been addressed. I would like to keep the score.

**Key Questions For Authors:**

1. Why not use autoregressive protein LMs, which would allow standard DPO to be applied directly without the union-mask workaround?
2. In Figure 5, why does performance saturate after 10K with CPT?
3. How well would the framework generalize to non-VHH antibodies?

**Limitations:**

Yes.

**Strengths And Weaknesses:**

**Strength**
1. The authors introduce Exp-1K and Camel-4M, two proprietary datasets that significantly exceed existing public benchmarks in scale.
2. The authors innovatively apply DPO to protein language models for antibody expression ranking, successfully improving performance significantly over previous methods.


**Weakness**
1. The test set contains only 126 sequences, which still might be too small to evaluate the model's performance.

---

> ### Author Rebuttal · Authors · 2026-03-31
>
> > **The test set contains only 126 sequences, which still might be too small to evaluate the model's performance. (Weakness)**
>
> We thank the reviewer for the constructive feedback. We appreciate the recognition in Strength 1 that our proprietary datasets, Exp-1K and Camel-4M, "significantly exceed existing public benchmarks in scale". Regarding the test set size, we would like to provide additional context:
> * Domain-Specific Constraints: In the field of antibody expression, high-fidelity experimental data is a well-known bottleneck due to the high cost and complexity of wet-lab validation. While 126 sequences might appear small by general ML standards, it constitutes 10% of our 1254 labeled sequences, a scale that is already at the top tier for this specific task.
> * Comparison with Existing Literature: Many baseline studies in this domain rely on significantly smaller datasets. For example, the g-DPO study evaluates expression on a total of only 76 sequences. Our Exp-1K (1254 sequences) is larger than the entire labeled datasets used in several previous works.
> * Statistical Robustness: Despite the inherent data scarcity in antibody research, our results demonstrate consistent performance gains across multiple backbones.
>
> > **Why not use autoregressive protein LMs? (Question 1)**
>
> We thank the reviewer for this insightful question regarding the architectural choice. We chose Masked Language Models (MLM) rather than Autoregressive (AR) models for three primary reasons:
>
> * State-of-the-Art (SOTA): The most powerful antibody and protein language models (e.g., AntiBERTa2, IgBERT) are predominantly based on MLM architectures. Utilizing these models allows our framework to leverage the most robust evolutionary representations currently available in the field.
> * Biological Inductive Bias: Protein evolution is a non-directional, co-evolutionary process. The properties of a specific residue are often determined by its spatial neighbors, which may be distant in the linear sequence. MLMs, with their bidirectional attention, are inherently better suited to capture these complex signals compared to AR models.
>
> > **Which would allow standard DPO to be applied directly without the union-mask workaround? (Question 1)**
>
> We adopted the union-masked approximation (as successfully demonstrated in g-DPO) because it provides a reliable gradient signal with only a single forward pass.
>
> > **In Figure 5, why does performance saturate after 10K with CPT? (Question 2)**
>
> The observed saturation after 10K sequences suggests that a subset of this size already captures the primary structural distribution and evolutionary patterns inherent in the Camel-4M library. This is consistent with the high data density and possible redundancy. However, we emphasize that the full 4M dataset consistently provides superior performance over the 10K sample (as shown in our clustering ablation). The massive scale of the full dataset is essential for capturing the "long tail" of the antibody landscape, which provides the incremental but critical gains observed in downstream tasks.
>
> > **How well would the framework generalize to non-VHH antibodies? (Question 3)**
>
> Theoretically, the algorithm has potential to apply to non-VHH antibodies if we have available data to train the model on VH-VL sequences.

---

> > ### Author Rebuttal · Reviewer_BN3p · 2026-04-03
> >
> > Thank you for the response. The author has addressed my concerns. I would like to keep my score.

---

### Official Review · Reviewer_nkFe · 2026-03-13

**Soundness:** 3
**Presentation:** 3
**Significance:** 3
**Originality:** 2
**Overall Recommendation:** 4
**Confidence:** 4

**Summary:**

This paper studies the problem of antibody expression ranking. The authors fine-tune pLM with CPT and DPO to predict antibody expression. The method consistently outperforms baselines on most metrics. An interesting finding is that preference learning can effectively learn from weak supervision.

**Compliance With Llm Reviewing Policy:**

Affirmed.

**Final Justification:**

The rebuttal addressed my concerns.

**Key Questions For Authors:**

- Will the dataset be publicly released?
- What's the main difference between ProteinDPO [1] and g-dpo [2]

[1] Aligning protein generative models with experimental fitness via Direct Preference Optimization
[2] g-dpo:Scalablepreference optimizationforproteinlanguagemodels.

**Limitations:**

See Weaknesses

**Strengths And Weaknesses:**

Strengths
- The problem of antibody expression ranking is important
- The paper is easy to follow.
- The method is effective in low-data constraints.

Weaknesses
- The novelty is limited. The main contribution of integrating scarce quantitative yields and large-scale weak positive supervision within a single training objective is similar to ProteinDPO and the union-masked log-likelihood approximation is adopted from g-DPO.
- The experiments are weak. Expression is one of the important developability. It is important to show the proposed methods can be transfered to other properties.

---

> ### Author Rebuttal · Authors · 2026-03-31
>
> > **Regarding novelty (Weakness 1, Question 2)**
>
> We thank the reviewer for the constructive feedback. Regarding the relationship between our work and existing methods, our paper identifies a specific **flaw in treating expression as a standard supervised learning task, as existing approaches often oversimplify the complex biological constraints involved, which distinguishes our approach from the existing literature, such as ProteinDPO and g-DPO**.
>
> While we build upon the mathematical foundations of g-DPO (union-masked log-likelihood) and ProteinDPO, our work is not a simple extension. ProteinDPO primarily focuses on aligning generative models with experimental fitness using direct labels, while g-DPO provides a scalable preference optimization framework. **However, both remain conceptually anchored within the standard supervised frameworks. Our core contribution is breaking away from this paradigm to solve the severe data scarcity in antibody expression.** We demonstrate how to transform massive, non-labeled immunization data (Camel-4M) into meaningful ranking signals to supervise the model when high-quality labels (Exp-1K) are insufficient for traditional training.
>
> Critically, our empirical results (Table 5) show that **directly applying these techniques to supervised data only is insufficient and can even be outperformed by simple MLP baselines.** This proves that the novelty and performance gains lie not in the algorithms themselves, but in the novel interaction between weak immunization signals and preference learning to solve the data-scarcity bottleneck in antibody design, where standard supervised methods, including direct applications of ProteinDPO or g-DPO, would struggle to perform effectively.
>
> > **Regarding experimental scope and property generalization (Weakness 2)**
>
> We emphasize that while our evaluation focuses on expression, the framework is fundamentally generalizable:
>
> * Methodological Generality: Our proposed framework, including the IMGT-based alignment and the union-masked log-likelihood approximation, is fundamentally a general-purpose solution for adapting masked protein language models to any sequence-level ranking task. It can be directly applied to other properties where pairwise is available.
> * The "Weak Supervision" Paradigm: A core contribution of this paper is demonstrating how large-scale weak positive supervision can mitigate the scarcity of labeled data. We identified antibody expression as the property to validate this paradigm because of the unique biological link between camel-derived immunization data and expressibility.
>
> > **Regarding data release (Question 1)**
>
> The data is proprietary. We plan to release the dataset upon completion of the patenting process.
>
> > **Importance of expressibility research**
>
> We would like to emphasize that **antibody expressibility is a critical developability constraint with direct implications for development cost and program risk**. Poorly expressed candidates introduce significant liability, requiring iterative redesign, increasing CMC burden, and delaying timelines. Accordingly, developability assessment is now recognized as an essential component of early-stage antibody discovery [1].
>
> This challenge is further amplified by recent advances in AI-driven protein and antibody generation. While these models enable rapid exploration of sequence space, they do not guarantee that generated sequences satisfy key developability constraints such as expression and stability, and still require substantial experimental validation [2, 3].
>
> As a result, expression prediction is not a secondary task but a critical gating factor for translating computational designs into viable therapeutic candidates. Our work directly addresses this gap by focusing on expression-aware ranking under realistic data scarcity.
>
> ref:
>
> [1] Zhang, Weijie, et al. "Developability assessment at early-stage discovery to enable development of antibody-derived therapeutics." Antibody Therapeutics 6.1 (2023): 13-29.
>
> [2] Cho, Yehlin, et al. "Stable de novo protein design via joint conformational landscape and sequence optimization." Nature Communications (2025).
>
> [3] Bennett, Nathaniel R., et al. "Atomically accurate de novo design of antibodies with RFdiffusion." Nature 649.8095 (2026): 183-193.

---

> > ### Author Rebuttal · Reviewer_nkFe · 2026-04-04
> >
> > My concerns have been adequately addressed. I raise my score to 4.

---

### Official Review · Reviewer_V7KG · 2026-03-13

**Soundness:** 3
**Presentation:** 2
**Significance:** 2
**Originality:** 3
**Overall Recommendation:** 3
**Confidence:** 5

**Summary:**

This paper focuses on the antibody expression ranking task optimization. A unified preference-based learning framework that integrates scarce quantitative expression data is proposed. Specifically, the authors first perform continual pretraining of the pretrained protein language model using a standard masked language modeling objective on the camel dataset. Subsequently,  DPO training procedure is executed to satisfy relative preferences between antibody sequences. Two-stage training based on ProtBERT, IgBERT, and AntiBERT2 consistently yields improvements in experimental performance.

**Compliance With Llm Reviewing Policy:**

Affirmed.

**Key Questions For Authors:**

Theoretically:

1. The antibody modeling domain often exhibits higher data redundancy relative to general protein modeling domain. Within the proposed   two-stage training pipeline, were data clustering procedures incorporated for Exp-1K and Camel-4M? Additionally, what impact does data clustering have on performance in downstream prediction tasks?

Technically:

2. The field of antibody modeling currently faces a scarcity of high-quality labeled data. Do the authors plan to open-source the Exp-1K and Camel-4M resources introduced in this study?
3. It would be beneficial if the authors could provide a richer set of evaluation results for protein and antibody foundation models on the antibody expression ranking task, thereby enabling a clearer assessment of the performance of the constructed method.

**Limitations:**

Please describe the potential societal impacts arising from the limitations of this research work.

**Strengths And Weaknesses:**

Strengths:
1. This paper focuses on the antibody expression ranking task, which a frontier scientific problem in protein modeling.
2. Building on three well-established protein language models (i.e., AntiBERT2, ProtBERT, IgBERT), the authors conduct a comprehensive experimental evaluation, documenting performance gains from the initial base models through the introduced masked language modeling (MLM) training phase and the subsequent Direct Preference Optimization (DPO) training phase. The experimental setup is clearly articulated, substantiating the advantages conferred by the proposed method.

Weaknesses:
1. LLM post-training techniques such as MLM and DPO have already been widely validated in prior studies for optimizing the performance of protein large language models. Consequently, this manuscript appears primarily as a translational application of mature methodologies and lacks sufficient theoretical novelty.
2. The experimental section adopts a two-stage training pipeline based on three base/backbone models to enhance performance, providing informative ablation references; however, it does not incorporate a sufficiently diverse set of base models, which limits a more accurate evaluation of the method’s actual performance level.

Minor:
1. As a research contribution in the AI4S domain, the manuscript is recommended to add an Impact Statement section to delineate the potential impacts.
2. The current Abstract section does not articulate the existing research background with sufficient clarity.

---

> ### Author Rebuttal · Authors · 2026-03-31
>
> > **Regarding novelty (Weakness 1)**
>
> Our paper identifies a specific **flaw in treating expression as a standard supervised learning task, as existing approaches often oversimplify the complex biological constraints involved**.
>
> While we agree that MLM and Preference Optimization are established in other contexts, our work is not a mere "translational application". Instead, we address a critical and underexplored problem setting: learning under highly heterogeneous supervision. Antibody modeling operates in a unique data regime where high-quality labels (Exp-1K) are extremely scarce due to wet-lab costs, while biologically relevant signals (Camel-4M) are abundant but weak.
>
> **Our core insight, developed through industrial collaboration, is that naturally occurring antibody molecules provide a latent resource of weak supervision** where expression levels are usually high, even without exact labels. Standard supervised methods fail to capture this. We demonstrate that Preference Optimization provides the necessary mathematical interface to unify these two disparate data sources.
>
> Critically, our empirical results (Table 5) show that **directly applying these techniques to supervised data only is insufficient and can even be outperformed by simple MLP baselines**. This proves that the novelty and performance gains lie not in the algorithms themselves, but in the novel interaction between weak immunization signals and preference learning to solve the data-scarcity bottleneck in antibody design.
>
>
> > **Regarding model diversity (Weakness 2, Question 3)**
>
> We appreciate the reviewer’s recognition in Strength 2 that our evaluation with three different protein and antibody protein foundation models is "comprehensive" and that the experimental setup "substantiates the advantages conferred by the proposed method". To further demonstrate generalizability, we include results using ESM-2 below.
> |Method|Stage I|AUC|MCC|Tau|SCC|Recall|Avg|
> |-|-|-|-|-|-|-|-|
> |Supervised|N|76.8|10.3|38.7|44.4|**46.2**|20.0|
> |Supervised|Y|77.5|25.4|37.9|44.2|42.3|31.0|
> |Ours (Stage II)|N|76.3|34.4|29.0|21.0|30.8|31.6|
> |Ours (Stage I+II)|Y|**81.0**|**38.9**|**49.5**|**44.5**|**46.2**|**43.9**|
>
> Results show that our framework outperforms the baselines.
>
>
> > Regarding data redundancy and clustering (Question 1)
>
> Exp-1K (1,254) is too small to require clustering. Camel-4M used NGS-based non-redundant VHH libraries (duplicates removed). In Stage I (continual pre-training), we learn the distribution of the data, where some level of redundancy is beneficial. For Stage II, we conducted an ablation study comparing the full Camel-4M against (1) a 10K random sample and (2) a 10K cluster-representative set (via MMSeqs2).
>
> |Data size|AUC|MCC|Tau|SCC|Recall|Avg|
> |-|-|-|-|-|-|-|
> |Full|82.4|43.7|46.3|50.8|46.2|45.0|
> |10K(MMSeqs2)|81.3|41.6|43.8|48.0|46.2|42.7|
> |10K(Random)|81.1|45.8|39.4|39.2|38.5|42.5|
>
> Results showed that performance for both 10K sets was similar and slightly lower than using the full dataset. This suggests that the scale of weak supervision is more critical for performance than redundancy reduction in this framework.
>
>
> > **Regarding data release (Question 1)**
>
> The data is proprietary. We plan to release the dataset upon completion of the patenting process.
>
>
> > **Others (Minor 1 & 2, limitations)**
>
> We will include the Impact Statement and refine the abstract/limitations in the camera-ready version.
>
> > **Importance of expressibility research**
>
> We would like to emphasize that **antibody expressibility is a critical developability constraint with direct implications for development cost and program risk**. Poorly expressed candidates introduce significant liability, requiring iterative redesign, increasing CMC burden, and delaying timelines. Accordingly, developability assessment is now recognized as an essential component of early-stage antibody discovery [1].
>
> This challenge is further amplified by recent advances in AI-driven protein and antibody generation. While these models enable rapid exploration of sequence space, they do not guarantee that generated sequences satisfy key developability constraints such as expression and stability, and still require substantial experimental validation [2, 3].
>
> As a result, expression prediction is not a secondary task but a critical gating factor for translating computational designs into viable therapeutic candidates. Our work directly addresses this gap by focusing on expression-aware ranking under realistic data scarcity.
>
>
> ref:
>
> [1] Zhang, Weijie, et al. "Developability assessment at early-stage discovery to enable development of antibody-derived therapeutics." Antibody Therapeutics 6.1 (2023): 13-29.
>
> [2] Cho, Yehlin, et al. "Stable de novo protein design via joint conformational landscape and sequence optimization." Nature Communications (2025).
>
> [3] Bennett, Nathaniel R., et al. "Atomically accurate de novo design of antibodies with RFdiffusion." Nature 649.8095 (2026): 183-193.

---

> > ### Author Rebuttal · Reviewer_V7KG · 2026-04-04
> >
> > Thank the authors for their detailed response. Specifically, the additional experiments on data clustering have addressed my concern, and the results suggest that, on a small-scale dataset, reducing redundancy does not lead to a performance improvement. On the other hand, based on my discussion with the authors, I still believe that the technical novelty of this work is limited; it appears to be more of a data-driven approach tailored to the specific characteristics of the antibody macromolecule domain. Moreover, the lack of assurance regarding open access to the dataset further diminishes the potential contribution of this paper to the broader research community. Taking these factors into consideration, I have decided to keep my score unchanged.

---

> > > ### Author Response · Authors · 2026-04-06
> > >
> > > > **TL;DR**: First, we argue that the method proposed here is novel by demonstrating the effectiveness of an under-appreciated form of weakly supervised learning which is rare in many datasets but very common in molecular biology, especially for larger mammals. Thus, this work could have very significant impacts. Second, our approach to sharing data is necessitated by the very standard industry constraints, we are committed to sharing the data as openly as possible once industrial expectations such as patents are completed.
> > >
> > > ---
> > > We understand that the remaining concerns center on (1) technical novelty, (2) “data-driven” approaches and generalizability, and (3) dataset availability and broader impact. We address these points below.
> > >
> > > > **(1) On technical novelty**
> > >
> > > As noted in our prior rebuttal, **our work identifies a specific flaw in treating antibody expression as a standard supervised learning task, which oversimplifies the problem and ignores weak supervision signals present in the data**. In particular, beyond the small high-quality experimental dataset (Exp-1K), large-scale weak biological signals (Camel-4M) are naturally available, yet standard supervised frameworks fail to leverage them.
> > >
> > > Critically, as shown in Table 5 and discussed in our prior rebuttal, **directly applying advanced methods including DPO within a standard supervised framework is insufficient, and can even underperform simple MLP baselines**. Here, we provide additional evidence for this using Antiberta2 for fair comparison:
> > >
> > > |Dataset|Method|Val SCC|Test SCC|
> > > |-|-|-|-|
> > > |Jain|DPO|61.7|-20.4|
> > > ||MLP|38.9|20.9|
> > > |Garbinski|DPO|45.0|39.4|
> > > ||MLP|54.5|41.8|
> > > |PROPHET-Ab|DPO|43.4|23.1|
> > > ||MLP|34.5|34.8|
> > >
> > > **This shows that greater algorithmic novelty isn’t the only path to better performance in real antibody development, since DPO exhibits stronger overfitting compared to MLP**. While DPO has shown strong results on large-scale protein datasets, it underperforms on small-scale antibody datasets. This also reflects a systematic gap between the AI community, where researchers focus on large-scale proteins and complex models, and real-world antibody development, where small-scale datasets and simple MLPs are standard practice.
> > >
> > > Consequently, contributions in this context cannot be judged solely by model novelty.
> > >
> > > > **(2) On “data-driven” approach and generalizability**
> > >
> > > The reviewer’s concern about “domain-specificity” overlooks the fact that our framework is designed for a universal data paradigm in antibody discovery: the coexistence of small-scale experimental labels (e.g., Exp-1K) and a large-scale immunization database (e.g., Camel-4M). This “one small, one large” structure is exciting as a new challenge for ML as it is not a very common one in standard ML research, yet it is in the very nature of antibody R&D, as seen even in antibody binding affinity benchmarks for SARS-CoV-2 [1]. **Our framework is generalizable to many other industrial workflows where weak supervision derived from large mammals is available**, which, as justified in Appendix D.3, serves as a reliable prior.
> > >
> > > Furthermore, antibody expression is not merely “one of many” properties. It is the fundamental gating factor for developability (ie. viability) in an organism. As supported by recent literature (Nature 2026, Nature Comm. 2025, noted in our previous rebuttal), even the most novel generative models remain clinically irrelevant if their outputs cannot be expressed. This can be seen as another example of the hallucination crisis but for the molecular biology domain. Thus, **a dedicated paradigm for expression is not a limitation, but a necessary advancement for the broader community to achieve real-world impact**.
> > >
> > > > **(3) On dataset availability and broader impact**
> > >
> > > We note that the methodological contribution is fully specified using standard components and is applicable to any other datasets that have similar supervision formulations.
> > >
> > > In current immunization workflows, around 90% of antibodies are generated using small mammals due to cost efficiency, but only 5–10% use large mammals like camelids. In small animals, we did not observe the same weak supervision signal. Our findings that weak supervision in camelid-derived datasets can lead to improvements in downstream expression ranking could shift immunization strategies toward larger mammals.
> > > This could influence the design of other early-stage antibody programs and alter the broader development landscape.
> > >
> > > Finally, the dataset is currently part of an ongoing therapeutic development effort, which imposes constraints on immediate open access. We commit to release the data after the patenting process is completed. This constraints is very common in industrial drug development, where data availability often follows a delayed release cycle.
> > >
> > > [1] Tsuruta, Hirofumi, et al. "A SARS-CoV-2 interaction dataset and VHH sequence corpus for antibody language models." NeurIPS 2024

---

### Decision · Program_Chairs · 2026-04-30

**Decision:**

Accept (regular)

**Comment:**

This paper proposes a preference-based learning framework to address the critical bottleneck of antibody expression ranking under data scarcity. By adapting Direct Preference Optimization (DPO) alongside masked language modeling, the authors integrate a small set of labeled experimental data with a massive dataset of camelid-derived antibody sequences. The reviewers highlighted the high practical significance of the work for real-world therapeutic design and praised the comprehensive empirical evaluations across multiple protein language model backbones, which consistently demonstrated the method's superiority over standard supervised baselines.

During the review process, reviewers raised valid concerns regarding the paper's algorithmic novelty (correctly noting its reliance on established DPO, MLM techniques) and the use of proprietary datasets that limit immediate reproducibility. In the rebuttal, the authors demonstrated that applying standard DPO to the small supervised dataset alone actually underperforms simple baselines, and showed that the success of the framework relies on a clever, domain-specific data formulation: utilizing the natural in vivo maturation of camelid antibodies as a latent "weak positive" signal for expressibility, which allows off-the-shelf preference learning to bypass severe data scarcity.

While one reviewer maintained a negative score, arguing that this remains an application-heavy, data-driven approach rather than a fundamental algorithmic breakthrough, the consensus recognized that effectively mapping biological heuristics to established ML frameworks to solve critical domain bottlenecks is highly valuable. Crucially, the authors have committed to releasing the proprietary Camel-4M and Exp-1K datasets upon the completion of ongoing patent processes. We want to strongly emphasize that this explicit commitment to open-source the data is a primary factor in the decision to accept this paper. Because the paper's core contribution relies on this specific data paradigm, the broader ML community must be able to access the data to reproduce and build upon these findings. We strongly urge the authors to honor this release timeline. Given the strong empirical validation, the practical utility of this weak-supervision data strategy for antibody design, and the authors' commitment to data sharing, the merits of the work outweigh its limitations. Therefore, we recommend acceptance.